# Antibacterial Activity of Manganese Dioxide Nanosheets by ROS-Mediated Pathways and Destroying Membrane Integrity

**DOI:** 10.3390/nano10081545

**Published:** 2020-08-07

**Authors:** Ting Du, Siya Chen, Jinyu Zhang, Tingting Li, Ping Li, Jifeng Liu, Xinjun Du, Shuo Wang

**Affiliations:** 1State Key Laboratory of Food Nutrition and Safety, College of Food Science and Engineering, Tianjin University of Science and Technology, Tianjin 300457, China; tingdu@tust.edu.cn (T.D.); siycmail@163.com (S.C.); jinyuzhang@mail.tust.edu.cn (J.Z.); zoelxx@tust.edu.cn (P.L.); liujifeng111@gmail.com (J.L.); 2Key Laboratory of Micro-Nano Materials for Energy Storage and Conversion of Henan Province, College of Advanced Materials and Energy, Institute of Surface Micro and Nano Materials, Xuchang University, Xuchang 461000, China; tingtingli101@xcu.edu.cn; 3Tianjin Key Laboratory of Food Science and Health, School of Medicine, Nankai University, Tianjin 300071, China

**Keywords:** manganese dioxide, nanosheets, *Salmonella*, antibacterial, destroy membrane integrity

## Abstract

Manganese dioxide (MnO_2_) nanosheets have shown exciting potential in nanomedicine because of their ultrathin thickness, large surface area, high near-infrared (NIR) absorbance and good biocompatibility. However, the effect of MnO_2_ nanosheets on bacteria is still unclear. In this study, MnO_2_ nanosheets were shown for the first time to possess highly efficient antibacterial activity by using *Salmonella* as a model pathogen. The growth curve and surface plate assay uncovered that 125 μg/mL MnO_2_ nanosheets could kill 99.2% of *Salmonella*, which was further verified by fluorescence-based live/dead staining measurement. Mechanism analysis indicated that MnO_2_ nanosheet treatment could dramatically induce reactive oxygen species production, increase ATPase activity and cause the leakage of electrolytes and protein contents, leading to bacterial death. These results uncover the previously undefined role of MnO_2_ nanosheets and provide novel strategies for developing antimicrobial agents.

## 1. Introduction

Bacterial infections have become one of the world’s largest public health problems; hundreds of thousands of people are reported to suffer from bacterial infections every year [1,2]. To solve this problem, antibacterial drugs are widely used and the abuse of traditional antibiotics leads to antibiotic resistance, making such infections extremely difficult to treat. Currently, nanomaterials have attracted increasing attention as new antibacterial agents owing to their high specific surface area and excellent physicochemical properties [3,4,5,6,7]. Up to now, nanomaterials that have been reported to possess a certain degree of antibacterial activity are mainly divided into two categories—(i) two dimensional (2D) materials (GO, rGO, MoS_2_, WS_2_) [8,9] as well as (ii) metal nanoparticles and metal oxide nanoparticles (AgNPs, ZnO, AuNRs, TiO_2_) [10,11,12]. Among them, 2D materials with a nanoscale thickness are considered as the most promising antibacterial materials.

Ultrathin MnO_2_ nanosheets, an attractive novel type of 2D material, have attracted widespread attention in energy storage, biological analysis, cell imaging and drug delivery thanks to their large surface area, high near-infrared (NIR) absorbance and good biocompatibility [13,14,15,16,17,18,19,20]. Two-dimensional MnO_2_ nanosheets exhibit unique photothermal conversion ability due to their inherently thinness, making them excellent photothermal conversion materials for highly effective photothermal therapy (PTT) anti-tumor [21]. MnO_2_ nanosheets possess a strong and broad optical absorption spectrum, which provides an high-efficiency wide-spectrum fluorescence quencher for the design of fluorescence opening probes to monitor transmission potency [22]. Additionally, MnO_2_ nanosheets can be reduced to Mn^2+^ by intracellular glutathione, providing activated magnetic resonance and fluorescence signals to monitor delivery efficiency [23,24]. More importantly, manganese is a crucial element in human body and its metabolism does not trigger a serious immune response. Despite these advances in nanomedical applications of MnO_2_-based materials, little is known about the potential impact of MnO_2_ nanosheets on bacteria so far.

The objectives of the current study were to investigate for the first time the anti-*Salmonella* activity of MnO_2_ nanosheets in vitro and explored the interaction mechanism of MnO_2_ nanosheets with it. *Salmonella*, a major cause of food-borne disease outbreaks, remains a serious problem in the poultry industry and the public health sector and it is important to develop new methods and new drug formulations to combat *Salmonella*. We synthesized MnO_2_ nanosheets by ultrasonic treatment at room temperature and studied their antibacterial ability by using *Salmonella* as a model pathogen. Our results suggest that MnO_2_ nanosheets have highly effective antibacterial activity and may be used as a substitute or supplement for antibiotics to control foodborne pathogens.

## 2. Materials and Methods 

### 2.1. Reagents and Materials 

Manganese chloride tetrahydrate (MnCl_2_·4H_2_O) and tetramethylammonium hydroxide were purchased from Shanghai Reagent Chemical Co. (Shanghai, China). Hydrogen peroxide (H_2_O_2_, 30 wt%) and other chemicals of analytical grade were obtained from Sinopharm Chemical Reagent Co., Ltd. (Shanghai, China). Dulbecco’s modified Eagle’s medium (DMEM), trypsin, fetal bovine serum (FBS), MTT and penicillin streptomycin were obtained from GIBCO Invitrogen Corp (Waltham, MA, USA); 2′,7′-dichlorofluorescin diacetate (DCFH-DA) from Sigma-Aldrich (St. Louis, MO, USA); propidium iodide (PI), 4′-6-diamidino-2-phenylindole (DAPI) and Coomassie brilliant blue G-250 kit products from Beijing Solarbio Science & Technology Co., Ltd. (Beijing, China). 

### 2.2. Preparation of MnO_2_ Nanosheets

The manganese dioxide nanosheet was synthesized as previously reported [23]. Briefly, 20 mL of a mixed aqueous solution containing 0.6 M tetramethylammonium hydroxide and 3wt% H_2_O_2_ was injected to 10 mL 0.3 M MnCl_2_ solution within 15 s under vigorous agitation. When mixed, the solution immediately turned dark brown, indicating that Mn^2+^ was oxidized to Mn^4+^. The obtained dark brown suspension was stirred vigorously overnight at room temperature and then the bulk manganese dioxide was centrifuged at 2000 rpm for 10 min, washed with deionized water and methanol, dried at 60 °C and placed at room temperature for subsequent use. To synthesize the MnO_2_ nanosheet, 20 mg bulk manganese dioxide was dispersed in 10 mL of deionized water, sonicated for 10 h, followed by centrifugation at 2000 rpm for 10 min, the supernatant was collected for later use. Finally, inductively coupled plasma mass spectrometry (ICP-MS 7500cx, Agilent, Santa Clara, CA, USA) was used to determine the concentration of prepared MnO_2_ nanosheets.

### 2.3. Antibacterial Assays

The plate count method and growth curves were used to test the activity of MnO_2_ nanosheets against *Salmonella*. Briefly, 10 μL of *Salmonella* was injected to 10 mL lysogeny broth (LB) liquid medium and then cultured in a 37 °C incubator until 0.6–0.8 optical density (OD). After centrifugation at 6000 rpm for 5 min, the bacterial cells were harvested and washed with phosphate buffer saline (PBS), followed by discarding the supernatant and resuspending the cells in PBS buffer.

Next, 100 μL (10^6^ CFU/mL bacteria) was incubated at 4 °C for 1 h with MnO_2_ nanosheets at different concentrations (500, 250, 125, 62.5, 31.3 μg/mL). Each group was divided into two tubes, one test tube was used for plate test and the other was used for growth curves test. After dilution and uniformly plating in LB solid medium, the treated bacteria were cultured at 37 °C for 12 h. Meanwhile, the bacteria exposed with PBS were used as control. The antibacterial activity of MnO_2_ nanosheets was evaluated by the survival rate or the colony-forming units (CFUs). The survival rate was calculated by the equation—survival rate (%) = CFU_(experimental group)_/CFU_(control group)_ × 100% [11].

The other tube mixtures were transferred into 10 mL LB broth. The *Salmonella* samples were then incubated at 37 °C in an incubator with constant agitation at 200 rpm. Equal of samples were taken every 1 h and the OD value (OD600) at 600 nm wavelength was measured with an ultraviolet spectrophotometer to plot the growth curve of *Salmonella*.

### 2.4. Live/Dead Staining Assay

The reliability of the above test was confirmed by fluorescence microscopic analysis of the cell survival rate after exposure to MnO_2_ nanosheets using a Live/Dead bacterial viability kit. We stained the bacterial suspension with 40,6-diamidino-2-phenylindole (DAPI) and PI fluorescent dye, respectively, because DAPI can rapidly penetrate the unruptured membrane of living cells and bind to DNA in the nucleus, whereas PI can solely penetrate the ruptured membrane and bind to double stranded DNA. In the logarithmic phase, the *Salmonella* cells were cultured in LB medium in the presence or absence of 125 μg/mL MnO_2_ nanosheets. After collection by centrifugation, the cells were stained first with propidium iodide (PI, 10 μg/mL) for 10 min and then with 4′-6-diamidino-2-phenylindole (DAPI, 5.0 μg/mL) for 5 min in the dark. After washing with PBS, the cells were observed with a fluorescence microscope.

### 2.5. Morphological Analysis of Bacterial Cells

The morphologies of *Salmonella* cells untreated and treated with MnO_2_ nanosheets were observed by scanning electron microscope (SEM) and transmission electron microscopy (TEM). Briefly, after exposure to MnO_2_ nanosheets for 2 h, the *Salmonella* cells (10^7^ CFU/mL) were centrifuged (6000 rpm, 5 min), washed with PBS and then fixed with 2.5% glutaraldehyde at room temperature for 4 h. Next, the cells were washed with PBS buffer, followed by successive dehydration for 15 min with ethanol at different concentrations (30, 50, 70, 90 and 100%) and then freeze-drying. Finally, the samples were characterized separately by SEM and TEM.

### 2.6. Leakage of Intracellular Components 

The integrity of the bacterial cell membrane can be reflected by the leakage of intracellular compounds. The extracellular protein concentration of the bacterial suspension was determined by using the Coomassie Brilliant Blue G-250 kit (Beijing Solarbio Science & Technology Co., Ltd. Beijing, China). Briefly, *Salmonella* cells (10^7^ PFU/mL) in logarithmic phase were unexposed or exposed with 125 μg/mL MnO_2_ nanosheets for 2 h at 4 °C, followed by centrifugation at 10,000 r/min for 5 min to collect the supernatant. Next, the treated bacterial solution (50 μL) was supplemented with 200 μL of the protein working solution in each well of the 96-well plate and then cultured at 37 °C for 5 min. Finally, the protein leakage was evaluated by measuring the absorbance at 562 nm using an Enzyme Linked Immunosorbent Assay (ELISA) microplate reader (Waltham, MA, USA).

### 2.7. Measurement of Electrolyte Leakage

Electrolyte leakage assay was carried out as reported by Steel et al. [25]. Briefly, *Salmonella* cells in logarithmic phase were unexposed or exposed with MnO_2_ nanosheets at different concentrations for 2 h at 4 °C, followed by centrifugation at 6000 r/min for 5 min to collect the supernatant. Finally, the electrical conductivity of the suspension was measured using a conductivity meter.

### 2.8. Reactive Oxygen Species (ROS) Assay

The intracellular ROS level was investigated by fluorescence imaging. Briefly, *Salmonella* cells in logarithmic phase were unexposed or exposed with 125 μg/mL MnO_2_ nanosheets for 2 h at 4 °C, followed by rinsing with PBS and resuspending the cells in PBS solution. After adding one hundred microliter of 2′,7′-dichlorodihydrofluorescein diacetate (DCFH-DA, 10 μM), the cell suspension was cultured for another 1 h in darkness, followed by washing the excess dye with PBS and observing the samples with a fluorescent microscope.

### 2.9. Analysis of Bacterial Total ATPase Activity

After exposure to MnO_2_ nanosheets and three washes in PBS, the *Salmonella* cells were resuspended in PBS buffer. The pure *Salmonella* not treated by the MnO_2_ nanosheets were used as blank control group. Next, the bacterial suspension was placed in constant cell disruption systems for bacterial fragmentation and then treated according to the instructions of the ATPase Kit (ATPase kit for cells; Jiancheng Biotechnology Co., Nanjing, China). Finally, the activity of ATPase was calculated by measuring the absorption value at 636 nm.

### 2.10. Cytotoxicity Assay 

The cytotoxicity of the MnO_2_ nanosheets was evaluated via 3-[4,5-dimethylthiazol-2-thiazolyl]-2,5-diphenyl tetrazolium bromide (MTT) assay [26]. Briefly, at 90–100% confluence, Vero cells were treated with various concentrations of MnO_2_ nanosheets (500, 250, 125, 62.5, 31.3 μg/mL) for 12 and 24 h, followed by supplementation with 20 μL MTT reagent (3-(4,5-dimethyl-2-thiazolyl)-2,5-diphenyl-2-Htetrazolium bromide) in each well and incubation for another 4 h. After removing the supernatant, the formazan crystals were dispersed in 150 μL/well dimethysulfoxide. Vero cells treated with the Dulbecco’s modified Eagle’s medium (DMEM) (2% foetal bovine serum (FBS)) were used as control. Finally, the absorbance was determined at 570 nm with an Enzyme Linked Immunosorbent Assay (ELISA) microplate reader and the cell survival rate was calculated.

### 2.11. Statistical Analysis

The experimental data were analyzed using an independent *t*-test or one-way ANOVA test. Data are shown as the mean ± SE. The Student’s *t* test was used to calculate the statistical significance between the control group and the experimental group. * and ** indicate values at *p* < 0.05 and *p* < 0.01, respectively.

## 3. Results and Discussion

### 3.1. Characterization of MnO_2_ Nanosheets

MnO_2_ nanosheets were fabricated via ultrasonicating bulk MnO_2_ as previously reported [23]. The surface morphology and optical properties of the MnO_2_ nanosheets were characterized via transmission electron microscopy (TEM), UV-vis adsorption spectrum and X-ray diffraction (XRD). The obtained MnO_2_ nanosheets had a broad adsorption in the range of 250–700 nm, with the strongest absorption peak at 364 nm, corresponding to the d-d band transition between the low energy (3d t_2g_) and high energy (3d e_g_) of manganese ions, which is caused by the ligand field of MnO_6_ octahedron in MnO_2_ lattices [27] (Figure 1B). In the TEM image, the typical 2D MnO_2_ nanosheets were shown to be successfully prepared (Figure 1A). In the XRD pattern (Figure 1D), there are two strong peaks at 36.7° and 65.7° 2θ [28], corresponding to analytical crystal diffraction planes of 100 and 110, indicating that the MnO_2_ was two-dimensional with six sides. Other diffraction peaks (001, 002, 003) reflected the preferred orientation of the crystal, further confirming that the manganese dioxide prepared has a polycrystalline sheet structure.

The characteristics of the obtained MnO_2_ nanosheets were further investigated by analyzing their surface functional groups and valence status with Fourier transform infrared (FTIR) spectroscopy and X-ray photoelectron spectroscopy (XPS). In Figure 1C, the characteristic peak at 506 cm^−^^1^ represented the symmetric stretching vibration Mn–O bond [27]. The band at 3388 cm^−1^ of MnO_2_ nanosheets was ascribed to the physically absorbed H_2_O and the stretching vibrations of O–H (N–H). The high-intensity peak at 1624 cm^−1^ was ascribed to the stretching vibration of C=O and the peaks at 1481, 1406, 948 and 761 cm^−1^ were caused by the bending vibration of C–H. The FTIR results suggested the presence of a large number of hydrophilic groups on the surface of MnO_2_ nanosheets. 

The full range XPS analysis of the MnO_2_ nanosheets displayed five evident peaks at 642.1, 521.1, 402.1, 287.1 and 198.1 eV, corresponding to Mn 2p, O 1s, N 1s, C 1s and Cl 2p, respectively (Figure 2A) [29,30]. The Mn 2p peaks at 653.7 and 642.3 eV were assigned to Mn2p1/2 and Mn2p3/2 (Figure 2B) [31]. The C 1s peak at 286.4 and 284.7 eV might indicate that carbon is mostly in the form of C–N–C and C–C (Figure 2D). The XPS spectrum of N 1s confirmed the presence of N-H (402.6 eV) bonds (Figure 2C). The XPS results were consistent with the FTIR results.

### 3.2. Antibacterial Activity of MnO_2_ Nanosheets

The antibacterial activity of the MnO_2_ nanosheets was quantitatively explored by analyzing the growth curves and survival rates of bacteria treated with MnO_2_ nanosheets using the plate count method and Salmonella as a model microorganism. In Figure 3B, the agar plate exposed to different concentrations of MnO_2_ nanosheets (31.3–500 μg/mL) showed only a few or no colonies in a dose-dependent manner versus the control (A, 0 μg/mL). Figure 3A shows the corresponding statistics (survival rate) for the antibacterial effect of each treatment. The death rate was >1% for the treatment of 62.5 μg/mL MnO_2_ nanosheets and reached over 99.9% for the treatment of 250 μg/mL. Therefore, the number of colonies on the plate and the survival rates of bacteria proved that MnO_2_ nanosheets do have excellent antibacterial activity against Salmonella replication.

Additionally, to verify whether MnO_2_ nanosheets had broad-spectrum antibacterial activity, we also studied its inhibitory effect on Staphylococcus aureus (Gram-positive) by the surface plate assay. The results were shown in Appendix A, a significant dose-dependent inhibitory effect was observed after exposure of different concentrations of MnO_2_ nanosheets. When the concentration of MnO_2_ nanosheet was 125 µg/mL, the survival rate of Staphylococcus aureus was only 6.26%. The high concentration of 250 µg/mL MnO_2_ nanosheets have a higher inhibitory effect on Staphylococcus aureus and the inhibition rate was up to 99.95%. These results indicated that MnO_2_ nanosheets also have antibacterial activity against Gram-positive bacteria.

The bacterial inhibitory effect of MnO_2_ nanosheets was further studied by measuring the growth kinetics of Salmonella cells in liquid medium. According to the cell suspension turbidity, the bacterial growth was tracked by testing the optical density (OD) at 600 nm. Apparently, increased concentrations of MnO_2_ nanosheets resulted in decreased bacterial survival versus the control (0 μg/mL), indicating a concentration-dependent suppression of the MnO_2_ nanosheets on the growth of Salmonella cells (Figure 4).

To further determine the authenticity of the CFU method, the live/dead bacterial staining method was used to further verify the above results. The results are shown in Figure 5, the untreated groups (A and B) showed weak red fluorescence and strong blue fluorescence, while a large number of Salmonella cells were seen to die from cell membrane rupture in the group (D) treated with 125 μg/mL MnO_2_ nanosheets and stained with PI. The results revealed that MnO_2_ nanosheets can destroy the bacterial membrane structure, which is not only consistent with the CFU results but also indicates the damage to the cell membrane. 

### 3.3. Cytotoxicity

Good cell compatibility is an important factor for MnO_2_ nanosheets as a potential antibacterial agent. The effect of MnO_2_ nanosheets on Vero cell viability was investigated by MTT assay. As shown in Figure 6, after incubation separately for 12 and 24 h, the viability of Vero cells was more than 90% in the treatments of 31.3–250 μg/mL MnO_2_ nanosheets relative to the control. At 125 μg/mL of MnO_2_ nanosheets, the cell viability was close to 94%. Additionally, at a high concentration of 500 μg/mL, MnO_2_ nanosheets showed almost no toxicity to cells after incubation for 24 h with a cell viability of approximately 81%. Fan et al. reported that the 60 μg/mL MnO_2_ nanosheets had low cytotoxicity to the tested cell lines [21]. Shi and co-workers found that after 24 h of incubation with 200 μg/mL MnO_2_ composites, hc-4T_1_ cells still showed more than 90% viability [32]. Our cytotoxicity results for MnO_2_ nanosheets on Vero cell lines were consistent with these reported results. Thus, the 125 μg/mL MnO_2_ nanosheet was used in the following mechanism research experiments.

### 3.4. Mechanism for Antibacterial Activity of MnO_2_ Nanosheets

Currently, nanoscale materials have become a new type of antibacterial agents due to their high specific surface area and unique chemical and physical properties. Many papers have mentioned that the bactericidal activity of nanomaterials is mainly attributed to physical damage (e.g., destruction of lipid molecules) and chemical damage (e.g., oxidative stress) [33,34,35,36,37,38,39,40]. In this study, a three-step approach was proposed to define the synergy of antibacterial activity—(1) Direct contact of MnO_2_ nanosheets with the bacterial membrane due to their special sheet-like structure; (2) Generation of reactive oxygen species; (3) Membrane damage, leakage of electrolytes and intracellular contents and decrease of ATPase activity, which contribute to bacterial death. 

#### 3.4.1. Salmonella Structure Observation by TEM and SEM after Incubation with MnO_2_ Nanosheets

Transmission electron microscopy (TEM) and scanning electron microscopy (SEM) are usually used to investigate the direct interactions between nanomaterials and biological cells [41]. In this study, the structural changes in the treated Salmonella were first observed by SEM. In the absence of MnO_2_ nanosheets, Salmonella had a typical rod shape with intact cell walls (Figure 7C). After exposure to MnO_2_ nanosheets, significant changes were detected in the morphology of Salmonella, with the cell wall being wrinkled and damaged and obvious variations in the cell shape and size (Figure 7D).

TEM experiments were performed to test whether MnO_2_ nanosheets can destroy the integrity of bacterial cell walls. In Figure 7A, the untreated Salmonella cells had a rod shape with an intact morphology. However, the treated bacterial cell wall was seriously damaged, coupled with leakage of contents and changes in cell length (Figure 7B). Cell elongation was reported as a typical bacterial response to stress, including exposure to bactericides [42,43]. Interestingly, in Figure 7B, the surface of the bacteria is seen to be covered and wrapped with MnO_2_ nanosheets, forming temporary and localized high-concentration aggregates near bacterial surface, resulting in the biological disconnection of cells from the surrounding environment and cell death. A possible explanation is that MnO_2_ nanosheets may directly cause physical damage to the bacterial membrane due to their special two-dimensional lamellar structure. Previous studies have shown that carbon nanotubes (CNTs) can encapsulate human intestinal bacteria and penetrate cell walls and cell membranes and the degree of damage is dependent on the diameter of the CNTs [44]. O. Akhavan et al. mentioned that the graphene (oxide) suspension can wrap E. coli in the aggregated flakes, causing the cells to be biologically separated from the surrounding environment and unable to proliferate and then the bacteria are permanently inactivated by near-infrared radiation [45]. Chen et al. reported that the effective antibacterial activity of GO is related to the unique monolayer structure [9]. The above results indicate that the MnO_2_ nanosheets possess strong antibacterial activity, which was well supported by the SEM results.

#### 3.4.2. Generation of ROS

Oxidative stress is considered as one of the important toxic mechanisms associated with the exposure of nanoparticles, which can interfere with the function of DNA and enzymes, thereby disrupting normal metabolism and killing bacteria [46]. The potential role of MnO_2_ nanosheets in inducing oxidative stress was evaluated by using the oxidation-sensitive fluorescent probe DCFH-DA to monitor ROS production in the presence of MnO_2_ nanosheets. This probe can passively diffuse into the cell through the cell membrane. In Figure 8, the treatment of MnO_2_ nanosheets was seen to display strong green fluorescent signals (Figure 8C) compared with the control Salmonella cells (Figure 8B). Meanwhile, pure MnO_2_ nanosheets and probe DCFH-DA incubation were set up as a negative control (Figure 8A). The absence of fluorescence indicated that the nanomaterial itself does not oxidize DCFH-DA, inferring that MnO_2_ nanosheets induce the production of intracellular ROS in bacteria, which might contribute to the antibacterial effect. ROS levels are reported to be related to antibacterial activity due to their destructive effect on bacterial cell membranes [47].

Oxidative stress occurs when cells are exposed to enhanced levels of ROS, such as free radicals, ·O_2_^−^, ·OH and H_2_O_2_ [48]. Then, the hydroxyl radical (·OH) and superoxide radical anion (·O_2_^−^) generated by MnO_2_ nanosheets without the presence of bacteria were detected using electron spin resonance (ESR) spin-trap (DMPO) method. From the Figure 8D,E, it can be seen that MnO_2_ nanosheets generated typical DMPO-·O_2_^−^ and DMPO-·OH signals but ·O_2_^−^ was the main species. Similar conclusions have also been reported by other antibacterial nanomaterials. Wang et al. prepared positively charged bis-quaternary ammonium salt (BQAS) and found that BQAS had a strong contact with bacterial cells and produced temporary high concentration reactive oxygen species (mainly ·O_2_^−^), which triggered oxidative stress and membrane damage in bacteria and played a bactericidal role [48]. Curcumin-modified AgNPs have been shown to have a strong antibacterial effect, mainly due to the increased production of reactive oxygen species (mainly superoxide production) and bacterial death caused by membrane damage [47].

#### 3.4.3. Perturbation of Membrane Integrity by MnO_2_ Nanosheets

Loss of Electrolytes. Direct contact between pathogens and MnO_2_ nanosheets indicated that the pathogens are wrapped by the MnO_2_ nanosheets, suggesting that MnO_2_ nanosheets may interfere with the plasma membrane integrity of the pathogens. To test this hypothesis, we investigated the conductivity of the bacterial solutions with or without various concentrations of MnO_2_ nanosheets. When the bacterial cell membrane structure is destroyed, the nutrients inside the bacteria will leak, including electrolytes, leading to the increase of conductivity in the bacterial solution environment. Therefore, monitoring the change of conductivity in the bacterial solution can predict whether the cell membrane structure is intact. In Figure 9A, the conductivity of the solution was shown to increase gradually with the increase of concentration in the MnO_2_ nanosheet-treated groups (31.3–250 μg/mL) versus the control group (0 μg/mL), demonstrating that MnO_2_ nanosheets could indeed disrupt the phospholipids of Salmonella membranes.

Decrease of ATPase Activity. The disrupted cell membrane of Salmonella was checked by examining its ATPase activity after exposure to MnO_2_ nanosheets. Total ATPase is a key enzyme present on the bacterial cell membrane, which is closely related to the bacterial metabolism. The results showed a decrease in the ATPase activity of Salmonella after exposure to 125 μg/mL MnO_2_ nanosheets (Figure 9B), further confirming that the structure of Salmonella is destroyed by MnO_2_ nanoparticles.

Loss of Protein Contents. Injuries were characterized by electrolyte leakage and increased protein content. Membrane damage has a great impact on the changes of membrane potential and membrane related energy conversion systems. The above conclusion was further confirmed by testing the leakage of protein contents. In Figure 9C, the level of protein contents was shown to be 5.0-fold higher in the bacterial solution treated with 125 μg/mL MnO_2_ nanosheets than in the untreated control. Zheng et al. designed the La@GO nanocomposites, which can achieve the destruction of antibiotic-resistant bacteria due to a unique extracellular multi-target invasion and killing mechanism, including lipid dephosphorylation, lipid peroxidation and peptidoglycan destruction [37].

All the above results demonstrated that the antibacterial activity of MnO_2_ nanosheets is attributed to the formation of ROS and the damage to the bacterial cell membrane, leading to bacterial death (Scheme 1). 

## 4. Conclusions

In this study, the MnO_2_ nanosheet was successfully synthesized using a simple ultrasonic method and characterized systemically by UV-vis, XRD, FTIR, XPS and TEM. The manganese dioxide was shown to be in a lamellar distribution. This is probably the first paper to characterize the superior antibacterial activity of MnO_2_ nanosheets against *Salmonella*. When exposed to 125 μg/mL MnO_2_ nanosheets, 99.2% of *Salmonella* could be killed. Potential antibacterial mechanism studies showed that MnO_2_ nanosheets could adsorb and encapsulate bacteria, forming provisional and local high-concentration aggregates on the surface of bacteria, leading to the production of more reactive oxygen species, the increase of ATPase activity, the leakage of electrolytes and protein contents and ultimately bacterial death. These results supply experimental evidence for further improving MnO_2_ nanosheets as a potential antibacterial agent. Further research should focus on the evaluation of the pharmacokinetics and cytotoxicity of MnO_2_ nanosheets using appropriate animal models.

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
