# Peer review of "Antibacterial Activity of Manganese Dioxide Nanosheets by ROS-Mediated Pathways and Destroying Membrane Integrity"

_nanomaterials, 2020, doi:10.3390/nano10081545_

Round 1

Reviewer 1 Report

Although the authors describe that the ROS play an important role for antibacterial performance of MnO2 nanosheets,  they do not mention about origin of the ROS produced on the surface of MnO2. Hence, please explain why were the ROS produced during incubation with the bacteria and what kind of ROS was produced (singlet oxygen, superoxide, hydroxyl radical?). 

Author Response

Responses to the comments of reviewers

Responses to Reviewer 1

Comment 1:

Although the authors describe that the ROS play an important role for antibacterial performance of MnO2 nanosheets, they do not mention about origin of the ROS produced on the surface of MnO2. Hence, please explain why were the ROS produced during incubation with the bacteria and what kind of ROS was produced (singlet oxygen, superoxide, hydroxyl radical?)

Response:

     Thanks for the scientific comments. ROS are normal and harmless byproducts in the aerobic metabolism process, but biological cells would cause excessive ROS generation under adverse conditions. A large number of previous studies have shown that the toxicity action mediated by oxidative stress on the biological cells commonly occurred under the treatment of the biological cells with various nanomaterials, such as metal oxide nanoparticles and carbon nanomaterials [ACS Appl. Mater. Interfaces 8 (2016) 24057; Materials Science & Engineering C 99 (2019) 255; Biomaterials 35 (2014) 4706; Biomaterials 30 (2009) 5979; ACS Nano 8 (2016) 24057; Coordination Chemistry Reviews 357 (2018) 1]. Chen et al. reported that graphene nanosheets (similar to the MnO2 nanosheets structure we studied) have high antibacterial activity, mainly due to its high functional group density and small size, which has more opportunities to interact with bacterial cells, resulting in cells deposition. Through direct contact, graphene nanosheets can induce membrane stress by destroying and destroying cell membranes, and then superoxide anion independent oxidation will occur [ACS Nano 5 (2011) 6971].

   Additionally, studies have shown that the antibacterial activity of nanomaterials is related the generation of ROS [Nano Lett. 19 (2019) 7645; Adv. Funct. Mater. (2018) 1802140 ]. Oxidative stress occurs when cells are exposed to enhanced levels of ROS, such as free radicals, •O2, •OH and H2O2 [ACS Omega. 3 (2018) 14517]. To verify what kind of ROS is generated, the hydroxyl radical (•OH) and superoxide radical anion (•O2) generated by MnO2 nanosheets without the presence of bacteria were detected using ESR spin-trap (DMPO) method. From the Figure 8D and 8E, it can be seen that MnO2 nanosheets generated typical DMPO-•O2and DMPO-•OH signals, but •O2was the main species.

Reviewer 2 Report

I suggest revising the title, it is too long.

Abstract:

Provide the meaning of NIR.

Keywords: I think manganese dioxide and nanosheets can be two separate keywords, also replace “destroy” with a more specific term

Line 41: provide the meaning of NIR

Line 42: thinness

Line 43: what is PTT?

Paragraph starting in line 54: please rewrite this paragraph; results should not be discussed in the introduction. Instead, state the aim of the study and mention the major measurements performed. Scheme 1 also belongs in the results section.

Methods:

Line 67: provide city, state (if US), and country

Line 70: provide city

Line 81: provide city, state (if US), and country

Line 85: provide the meaning of LB

Line 118: citation is missing; also “briefly”

Line 148: this section should be renamed “Results and discussion”

Line 152: the meaning of TEM should be given at the first mention (line 103); same for SEM

Line 190: do you mean number of CFU?

Figure 3 clearly shows inhibition of bacteria; however, the curves of Figure 4 are similar and only a moderate decrease in bacteria is seen even for the highest concentration. How can you explain this?

Figure 6: are the values significantly different? Please add to the figure.

Line 238: more citations should be added.

The discussion still needs some improvement, please add a few more citations. Comparison with previous publications would be particularly useful.

Author Response

Responses to Reviewer 2

Comment 1:

I suggest revising the title, it is too long.

Response:

Thanks for the suggestion. We revised the title in the revised manuscript.

Comment 2:

Provide the meaning of NIR.

Response:

Thanks for the suggestion. NIR is short for near-infrared. We have added the explanation when they were firstly used in the revised manuscript.

Position in

the original manuscript

Original manuscript

Revised manuscript

Line 16

high NIR absorbance

high near-infrared(NIR) absorbance

Comment 3:

Keywords: I think manganese dioxide and nanosheets can be two separate keywords, also replace “destroy” with a more specific term.

Response:

According to the reviewer’s suggestions, we have divided “manganese dioxide nanosheets” into two keywords and changed "destroy" to "destroy membrane integrity".

Position in

the original manuscript

Original manuscript

Revised manuscript

Line 26

Manganese dioxide nanosheets;

Manganese dioxide; nanosheets;

Line 26

destroy

destroy membrane integrity

Comment 4:

(1) Line 41: provide the meaning of NIR.

(2) Line 42: thinness

(3) Line 43: what is PTT?

Response:

(1) NIR is short for near-infrared. We have added the explanation when they were firstly used in the revised manuscript.

(2) We have corrected "thin thickness " for "thinness" and as suggested.

(3) PTT is short for photothermal therapy. We have added the explanation when they were firstly used in the revised manuscript.

Position in

the original manuscript

Original manuscript

Revised manuscript

Line 41

high NIR absorbance

high near-infrared (NIR) absorbance

Line 42

inherently thin thickness

inherently thinness

Line 43

effective PTT anti-tumor

effective photothermal therapy (PTT) anti-tumor

Comment 5:

Paragraph starting in line 54: please rewrite this paragraph; results should not be discussed in the introduction. Instead, state the aim of the study and mention the major measurements performed. Scheme 1 also belongs in the results section.

Response:

Following the suggestion, we revised this paragraph on line 54 in the revised manuscript.

Position in

the original manuscript

Original manuscript

Revised manuscript

Line 52-57

Here, we synthesized MnO2 nanosheets by ultrasonic treatment at room temperature and studied their antibacterial ability and related mechanisms by using Salmonella as a model pathogen. Bacterial survival analysis showed that the MnO2 nanosheets could kill 99.2% of Salmonella at a concentration of 125 μg/mL. Antibacterial mechanism analysis revealed the intertwining of bacteria and MnO2 nanosheets to form local aggregates, which destroyed the integrity of the cell membrane and caused bacterial death (Scheme 1).

The objectives of the current study were to investigate for the first time the anti-Salmonella activity of MnO2 nanosheets in vitro and explored the interaction mechanism of MnO2 nanosheets with it. Salmonella, a major cause of food-borne disease outbreaks, remains a serious problem in the poultry industry and the public health sector, and it is important to develop new methods and new drug formulations to combat Salmonella. We synthesized MnO2 nanosheets by ultrasonic treatment at room temperature and studied their antibacterial ability by using Salmonella as a model pathogen. Our results suggest that MnO2 nanosheets have highly effective antibacterial activity and may be used as a substitute or supplement for antibiotics to control foodborne pathogens.

Comment 6:

(1) Line 67: provide city, state (if US), and country; Line 70: provide city; Line 81: provide city, state (if US), and country.

(2) Line 85: provide the meaning of LB;

(3) Line 118: citation is missing; also “briefly”;

(4) Line 148: this section should be renamed “Results and discussion”.

Response:

Thanks for the suggestion.

(1) We added the detailed address as suggested in the revised manuscript.

(2) LB is short for lysogeny broth. We have added the explanation when they were firstly used in the revised manuscript.

(3) We added the citation in the revised manuscript.

(4) We have revised "Results " to "Results and discussion" as suggested.

Position in

the original manuscript

Original manuscript

Revised manuscript

Line 67

GIBCO Invitrogen Corp

GIBCO Invitrogen Corp (Waltham, MA, USA)

Line 70

Beijing Solarbio Science & Technology Co.,Ltd (China)

Beijing Solarbio Science & Technology Co.,Ltd (Beijing, China).

Line 81

(ICP-MS 7500cx, Agilent)

(ICP-MS 7500cx, Agilent, Santa Clara, California, USA)

Line 85

10 mL LB liquid medium

10 mL lysogeny broth (LB) liquid medium

Line 118

Steel et al

Steel et al [25]

Line 148

Results

Results and discussion

Comment 7:

(1) Line 152: the meaning of TEM should be given at the first mention (line 103); same for SEM;

(2) Line 190: do you mean number of CFU?

Response:

Thanks for the suggestion.

(1) SEM is short for scanning electron microscope. TEM is short for transmission electron microscopy. We have added the explanation when they were firstly used in the revised manuscript.

(2) “the number of plaques” means “number of colonies on the plate”. We have modified the expression in the revised manuscript.

Position in

the original manuscript

Original manuscript

Revised manuscript

Line 103

by SEM and TEM

by scanning electron microscope (SEM) and transmission electron microscopy (TEM)

Line 190

the number of plaques

the number of colonies on the plate

Comment 8:

Figure 3 clearly shows inhibition of bacteria; however, the curves of Figure 4 are similar and only a moderate decrease in bacteria is seen even for the highest concentration. How can you explain this?

Response:

Thank the reviewer for the scientific comments. Growth curve and plate count assay are two methods to characterize the antibacterial properties of materials. The growth curve of Figure 4 was used to study the dynamics of bacterial growth, and evaluate the antibacterial properties of MnO2 nanosheets. Besides the growth inhibition, microbicidal activity is an important indicator for antibacterial materials. The plate count assay of Figure 3 was carried out to study the bactericidal effects of the MnO2 nanosheets. Figures 4 and Figure 3 are different in the presentation of specific data, but the conclusions are consistent, proving that MnO2 nanosheets have strong antibacterial activity. This phenomenon is also reflected in other related studies [Mater. Sci. Eng. C 99 (2019) 255-263; ACS Appl. Mater. Interfaces 11 (2019) 32659-32669; Nano Lett. 19 (2019) 7645-7654].

Comment 9:

Figure 6: are the values significantly different? Please add to the figure.

Response:

Thanks for the suggestion. When the cells were exposed to 500 μg/mL MnO2 nanosheets for 12 h and 24 h, there was a significant difference in the cell survival rate compared with the control group. And we have revised Figure 6 in the revised manuscript.

Figure 6. Cytotoxicity of different concentrations of MnO2 nanosheets via MTT assay. Vero cells were incubated with MnO2 nanosheets separately for 12 and 24 h. Statistical significance was decided with a *p < 0.05.

Comment 10:

Line 238: more citations should be added.

Response:

Thanks for the suggestion. We added the citation of relevant research literature in  Line 238 in the revised manuscript.

Position in

the original manuscript

Original manuscript

Revised manuscript

Line 238

chemical damage (e.g., oxidative stress) [32]

chemical damage (e.g., oxidative stress) [33-40]

Comment 11:

The discussion still needs some improvement, please add a few more citations. Comparison with previous publications would be particularly useful.

Response:

Thanks for the suggestions. We have revised the “Results and discussion” section in the revised manuscript according to the reviewer’s suggestion.

Reviewer 3 Report

The authors assessed the antibacterial and biological activity of MnO2 nanosheets. The methodology is generally clear, but a few improvements are needed. The results are a bit weirdly presented, with some paragraphs containing discussions, while other no not.

Some acronyms have to be defined at their first occurrence (NIR, PTT).

Why was Salmonella chosen as representative for testing the antibacterial effect? A Gram-positive representative would have been welcome to be assessed in parallel.

Line 80 - centrifugation for how many minutes?

Chapter 2.3 describe the antibacterial effect (MIC / MBC). The growth curve methodology is missing, though is mentioned in line 84 and 197-199

Chapter 2.4 - mention the expected result after PI/DAPI staining here, rather than in lines 206-210.

2.6, 2.7, 2.8, 2.9, 2.10 - The control (untreated) is not mentioned.

Salmonella , E. coli, - should be written with italics allover the manuscript.

Overall, a good and complex study which prove the importance of new materials with antibacterial effects.

Author Response

Responses to Reviewer 3

Comment 1:

The authors assessed the antibacterial and biological activity of MnO2 nanosheets. The methodology is generally clear, but a few improvements are needed. The results are a bit weirdly presented, with some paragraphs containing discussions, while other no not.

Response:

Thanks for the suggestions. We have revised the “Results and discussion” section in the revised manuscript according to the reviewer’s suggestion.

Comment 2:

Some acronyms have to be defined at their first occurrence (NIR, PTT).

Response:

We carefully checked all the abbreviations, and added the explanation when they were firstly used in the manuscript.

Position in

the original manuscript

Original manuscript

Revised manuscript

Line 16

high NIR absorbance

high near-infrared (NIR) absorbance

Line 43

highly effective PTT anti-tumor

highly effective photothermal therapy (PTT) anti-tumor

Comment 3:                                                                   

Why was Salmonella chosen as representative for testing the antibacterial effect? A Gram-positive representative would have been welcome to be assessed in parallel.

Response:

We appreciate the reviewer for this constructive comment. Salmonella, a member of the family Enterobacteriaceae, is a gramnegative bacterium [Preventive Medicine 29 (2002) 400]. Salmonella is a major cause of foodborne disease outbreaks [Foodborne Bacterial Pathogens 1918 (2019) 3], with salmonellosis ranking highest in symptomatic food poisoning infections throughout the year [Veterinary Research Forum 11 (2020) 67]. In its onset in animals, Salmonella can invade the blood circulatory system through the intestinal lymph nodes, thereby infecting the whole body. Control of these foodborne enteric pathogens is a real challenge for the food industry and public health agency. Moreover, it is very difficult to protect the safety of food chains due to the resurgence of multidrug-resistant strains of foodborne pathogens. Therefore, the detection and effective prevention of Salmonella is of great importance to prevent its spread, and has attracted wide attention in the scientific community [Food Chemistry 324 (2020) 126859; Food Chemistry 325 (2020) 126868; Food Microbiology 91 (2020) 103543; Chinese Medical Innovation 6 (2009) 194; Journal of Clinical Medicine (2010) 115]. So we chose Salmonella as representative for testing the antibacterial effect.

Additionally, according to the reviewer’s suggestions, we investigated whether the MnO2 nanosheets can inhibit the reproduction of Staphylococcus aureus (Gram-positive) by the surface plate assay. The results were shown in Figure S1, a significant dose-dependent inhibitory effect was observed after exposure of different concentrations of MnO2 nanosheets. When the concentration of MnO2 nanosheet was 125 µg/mL, the survival rate of Staphylococcus aureus was only 6.26%. The high concentration of 250 µg/mL MnO2 nanosheets have a higher inhibitory effect on Staphylococcus aureus, and the inhibition rate was up to 99.95%. These results indicated that MnO2 nanosheets also have antibacterial activity against Gram-positive bacteria.

Comment 4:

Line 80-centrifugation for how many minutes?

Response:

Thanks for the suggestion. The centrifugation time on Line 80 is 10 minutes.

Position in

the original manuscript

Original manuscript

Revised manuscript

Line 80

followed by centrifugation at 2000 rpm

followed by centrifugation at 2000 rpm for 10 min

Comment 5:

Chapter 2.3 describe the antibacterial effect (MIC/MBC). The growth curve methodology is missing, though is mentioned in line 84 and 197-199.

Response:

Thanks for the suggestion. We have added the detailed method steps of the growth curve experiment in the revised manuscript.

The detailed steps are as follows:

(1) Briefly, 10 μL of Salmonella was injected to 10 mL LB liquid medium, and  then cultured in a 37 ℃ incubator until 0.6-0.8 optical density (OD). After centrifugation at 6000 rpm for 5 min, the bacterial cells were harvested and washed with PBS, followed by discarding the supernatant and resuspending the cells in PBS buffer.

(2) Next, 100 μL (106 CFU/mL bacteria) was incubated at 4 ℃ for 1 h with MnO2 nanosheets at different concentrations (500, 250, 125, 62.5, 31.3 μg/mL). Each group was divided into two tubes, one test tube was used for plate test, and the other was used for growth curves test.

(3) The other tube mixtures were transferred into 10 mL LB broth. The Salmonella samples were then incubated at 37 °C in an incubator with constant agitation at 200 rpm. Equal of samples were taken every 1 hour, and the OD value (OD600) at 600 nm wavelength was measured with an ultraviolet spectrophotometer to plot the growth curve of Salmonella.

Comment 6:

Chapter 2.4-mention the expected result after PI/DAPI staining here, rather than in lines 206-210.

Response:

Following the suggestion, we have revised the expected result after PI/DAPI dyeing mentioned in lines 206-210 to Chapter 2.4.

Position in

the original manuscript

Original manuscript

Revised manuscript

Line 206-210

“The reliability of the above test was confirmed by fluorescence microscopic analysis of the cell survival rate after exposure to MnO2 nanosheets using a Live/Dead bacterial viability kit. We stained the bacterial suspension with DAPI and PI fluorescent dye, respectively, because DAPI can rapidly penetrate the unruptured membrane of living cells and bind to DNA in the nucleus, whereas PI can solely penetrate the ruptured membrane and bind to double stranded DNA”

Changed to “To further determine the authenticity of the CFU method, the live/dead bacterial staining method was used to further verify the above results. The results are shown in Figure 5

Line 95 (Chapter 2.4)

<Inserted>

The reliability of the above test was confirmed by fluorescence microscopic analysis of the cell survival rate after exposure to MnO2 nanosheets using a Live/Dead bacterial viability kit. We stained the bacterial suspension with DAPI and PI fluorescent dye, respectively, because DAPI can rapidly penetrate the unruptured membrane of living cells and bind to DNA in the nucleus, whereas PI can solely penetrate the ruptured membrane and bind to double stranded DNA”.

Comment 7:

2.6, 2.7, 2.8, 2.9, 2.10 - The control (untreated) is not mentioned.

Response:

Following the suggestion, we have added the description of the control group in the revised manuscript.

Position in

the original manuscript

Original manuscript

Revised manuscript

Line 112 (Chapter 2.6)

logarithmic phase were incubated with

logarithmic phase were unexposed or exposed with

Line 119 (Chapter 2.7)

logarithmic phase were cultured with

logarithmic phase were unexposed or exposed with

Line 124

(Chapter 2.8)

logarithmic phase were incubated with

logarithmic phase were unexposed or exposed

Line 131 (Chapter 2.9)

the Salmonella cells were resuspended in PBS buffer. Next,

the Salmonella cells were resuspended in PBS buffer. The pure Salmonella not treated by the MnO2 nanosheets were used as blank control group. Next,

Line 141 (Chapter 2.10)

the formazan crystals were dispersed in 150 μL/well dimethysulfoxide. Finally,

the formazan crystals were dispersed in 150 μL/well dimethysulfoxide. Vero cells treated with the DMEM (2% FBS) were used as control. Finally,

Comment 8:

Salmonella , E. coli, - should be written with italics allover the manuscript.

Response:

Thanks for the suggestion. We carefully checked all the italics and revised in the revised manuscript.
